# Ecdysteroid Biosynthesis Halloween Gene *Spook* Plays an Important Role in the Oviposition Process of Spider Mite, *Tetranychus urticae*

**DOI:** 10.3390/ijms241914797

**Published:** 2023-09-30

**Authors:** Liang Wang, Zhuo Li, Tianci Yi, Gang Li, Guy Smagghe, Daochao Jin

**Affiliations:** 1Institute of Entomology, Guizhou University, Guiyang 550025, China; iq1312467wl@163.com (L.W.); li159753chen2022@163.com (Z.L.); yitianci@msn.com (T.Y.); guysma9@gmail.com (G.S.); 2Guizhou Provincial Key Laboratory for Agricultural Pest Management of the Mountainous Region, Guiyang 550025, China

**Keywords:** *Tetranychus urticae*, ecdysteroid, *Spook*, RNAi, oviposition

## Abstract

In insects, the ecdysteroid hormone regulates development and reproduction. However, its function in the reproduction process of spider mites is still unclear. In this study, we investigated the effect of the Halloween gene *Spook* on the oviposition of the reproduction process in a spider mite, *Tetranychus urticae*. The expression patterns of the ecdysteroid biosynthesis and signaling pathway genes, as analyzed by RT-qPCR, showed that the expression pattern of the Halloween genes was similar to the oviposition pattern of the female mite and the expression patterns of the vitellogenesis-related genes *TuVg* and *TuVgR*, suggesting that the Halloween genes are involved in the oviposition of spider mites. To investigate the function of the ecdysteroid hormone on the oviposition of the reproduction process, we carried out an RNAi assay against the Halloween gene *Spook* by injection in female mites. Effective silencing of *TuSpo* led to a significant reduction of oviposition. In summary, these results provide an initial study on the effect of Halloween genes on the reproduction in *T. urticae* and may be a foundation for a new strategy to control spider mites.

## 1. Introduction

The strong reproductive ability of insects allows them to expand the population in a short period, causing severe harm to crops. The reproductive process is regulated by steroid hormones. Ecdysteroid is the main form of insect steroid hormones [1,2] and plays crucial roles in ovarian development processes such as follicle cell formation, vitellogenin production stimulation, ovarian growth, and oocyte maturation [3,4,5,6,7]. Moreover, ecdysteroid could regulate egg production and the expression of oogenic genes in the reproductive process of insects [5,8].

In insects, the ecdysteroid hormone, 20-hydroxyecdysone (20E), is the active form of ecdysteroid is synthesized from cholesterol by a series of enzymes encoded by the *Halloween* genes, including *Spook* (*Spo*, *CYP307A1*), *Phantom* (*Phm*, *CYP306A1*), *Disembodied* (*Dib*, *CYP302A1*), *Shadow* (*Sad*, *CYP315A1*), and *Shade* (*Shd*, *CYP314A1*) [9,10,11,12,13,14,15,16,17,18]. Then, the synthesized 20E binds to its heterodimer receptor consisting of the ecdysteroid receptor (EcR) and the retinoid X receptor (RXR) [19,20,21,22] to initiate the transcriptional cascade of the downstream genes, including the early genes *E74* and *E75*, the early-late genes *HR3*, *HR4*, *E78*, and *Kr-h1*, the late genes *FTZ-f1* [15,23,24,25].

The ecdysteroid biosynthesis Halloween genes play an important role in insect reproduction. For instance, in the desert locust, *Schistocerca gregaria*, RNA interference-mediated knockdown of the *Halloween* genes induced a significant impact on oocyte development, oviposition, and the hatching of eggs [26,27]. In the supermodel insect, the fruit fly, *Drosophila melanogaster*, silencing *Halloween* genes led to a reduction in oviposition [28]. Similar results were also obtained in *Diaphorina citri* [29]. In the soft tick, *Ornithodoros moubata*, *Spook* expression was determined in the ovary from both final instar nymphs and adult females, and *shade* expression was frequently surged after engorgement and copulation in adult females, suggesting *Spook* and *shade* are involved in ovary development [30]. Similar expression was also detected in the Varroa mite, *Varroa destructor* [31], and in the migratory locust, *Locusta migratoria* [32]. In the rice planthopper, *Nilaparvata lugens*, silencing the expression of the *Spookier* led to less oviposition [8]. However, the function of Halloween genes on the reproduction process in spider mites is still unknown.

In Acari, it has previously been reported that the ecdysteroid hormone regulates *Vg* synthesis in the ixodid tick, *Amblyomma hebraeum* [33], and egg development in the American dog tick, *Dermacentor variabilis* [34]. However, there is no report about the regulatory function of the ecdysteroid hormone in the reproduction process of spider mites. *Tetranychus urticae* is an important pest mite in agriculture and can quickly develop resistance to chemical acaricides, in part due to its strong reproduction capacity [35,36].

In this study, we investigated the ecdysteroid biosynthesis and signaling pathway genes in relation to the oviposition of *T. urticae.* In addition, we performed an RNAi bioassay to illustrate the regulatory function of the ecdysteroid biosynthesis Halloween gene *Spook* in the reproduction process of the spider mite. We believe that our results can provide new insights into the reproduction regulatory mechanism in mites, and they may potentially be a foundation for a new strategy to control these important pest organisms in agriculture.

## 2. Results

### 2.1. Expression Dynamics of the Halloween Genes in the Oviposition Period

To investigate the whole oviposition process of the female mite in *T. urticae*, we followed the oviposition from the day 1 of the adult stage to death of the female mite (Appendix A). The result showed that there was a significant difference within these 10 days, which could be sorted into two stages: stage I (day 1–7) and stage II (day 8–10) (Figure 1). In stage I, the average daily egg production increased and reached an amount of about 7–8 eggs per day per mite on day 3–4, then followed by a plateau of 5–6 eggs per day per mite between day 5–7. In stage II, the average daily egg production showed a significant drop to 2–3 eggs per day per mite starting on day 8, and this remained until day 10 (Figure 1).

To obtain a deeper insight on the regulatory function of ecdysteroid-related genes in the oviposition process of female mites in *T. urticae*, we collected female mite samples on a daily basis from day 1 of the adult stage up to the death of the female at day 10. The transcript pattern of the ecdysteroid biosynthesis and signaling pathway genes and the vitellogenesis-related genes were determined by RT-qPCR. For the ecdysteroid biosynthesis Halloween genes, it was interesting that their expression showed a similar pattern as compared to the oviposition dynamics. In detail, the expression of *TuSpo*, *TuDib*, *TuSad*, and *TuShd* showed an increase during the first 2 days, and then, during the next days (between day 2–6), there were higher levels, but on day 7 there was a dramatic, significant drop (Figure 2A–D). For the ecdysteroid signaling genes, the expression pattern of *TuHR3* and *TuHR4* showed an increase during the first days with peak levels between day 4–6, and then there was a dramatic drop on day 8 that remained up to day 10 (Figure 2J,K). The expression level of *TuRXR1* also increased during the first 3 days, but then decreased gradually to a low level at day 7, and then it increased again (Figure 2F). For *TuRXR2* (Figure 2G), the pattern was stable the first 5 days then showed a peak on day 6, and this was followed by lower levels between day 7 and 10. In contrast, the expression levels of *TuEcR*, *TuE78*, E75, and *TuFtz-f1* showed a more constant pattern (Figure 2E,H,I,L); however, for *TuE75* and *TuFtz-f1*, their stable expression pattern showed a dip on day 2 and day 7, respectively (Figure 2H,L).

To investigate the expression pattern of the vitellogenin genes and their receptor gene in the reproductive process, the expression levels of *TuVg1*, *TuVg2*, *TuVg3*, and *TuVgR1* were measured in the female mite samples from day 1 of the adult stage up to the death of the female at day 10, as we had used for the ecdysteroid biosynthesis and signaling pathway genes (see Figure 2). The expression patterns of *TuVg1*, *TuVg2*, *TuVg3*, and *TuVgR1* showed an increase between day 1 and day 3–4 (Figure 3A–D), and then at day 7 there was a dramatic drop. Further on, between day 7–10, *TuVg1* and *TuVgR1* showed a small increase, while the *TuVg2* and *TuVg3* levels stayed constant at a low level. Interestingly, these expression patterns were similar to those of the Halloween genes and the oviposition dynamics. Our data showed that the ecdysteroid biosynthesis genes have a positive correlation with the vitellogenin genes and their receptor gene, suggesting that the ecdysteroid biosynthesis pathway genes are involved in regulating the vitellogenesis process of *T. urticae* reproduction.

### 2.2. RNAi of Halloween Gene Spook Reduced the Female Oviposition of T. urticae

To evaluate the regulation function of the ecdysteroid hormone on the oviposition of female mites in *T. urticae*, RNAi of the Halloween gene *Spook* was performed based on the positive relationship between the expression pattern of this ecdysteroid biosynthesis gene (Figure 2A) and the oviposition dynamics (Figure 1). The silencing efficiency against *TuSpo* at 24 h after injection of ds*TuSpo* was 64% (*p* < 0.0001) compared to the control group injected with ds*egfp* (Figure 4A). For the RNAi-female mites, the total number of eggs laid per female mite was significantly reduced by 69% (*p* = 0.0017) compared with the ds*egfp*-control mites (Figure 4C). In detail, the average daily number of eggs in the control group showed an increase between day 2–3 and day 8, while the oviposition in the RNAi-female mites was significantly reduced in this period by about 50–80% based on average numbers (Figure 4B). In addition, we analyzed the oviposition period in ds*TuSpo*-treated female mites and found that it was significantly shorter than in the ds*egfp*-control (*p* = 0.0001) (Figure 4D). Similarly, the lifespan of the ds*TuSpo*-females was significantly shorter (*p* < 0.0001) (Appendix A). These results indicate that Halloween genes are involved in the oviposition regulation of *T. urticae*. But there were no effects of the RNAi treatment (after injection of 60–80 ng of ds*TuSpo* per female mite) in the offspring of the surviving females (Appendix A).

### 2.3. Effect of Halloween Gene Spook Silencing on the Transcript Levels of Other Genes Downstream

In this part, we investigated the RNAi effect of *TuSpo* on the expression of other ecdysteroid biosynthesis and pathway genes and also on that of the vitellogenesis-related genes by use of RT-qPCR. The expression levels of *TuDib* and *TuSad* showed no difference in the ds*TuSpo*-injected group, while that of *TuShd* was reduced by 22% (*p* = 0.0195) (Figure 5A). Similarly, the expression of the ecdysteroid receptor *TuEcR* was reduced by 38% (*p* = 0.0197), while there were no differences for *TuRXR1* and *TuRXR2* (Figure 5B). Also, for the ecdysteroid signaling pathway genes, there is no difference in the expression level of *TuHR4*, *TuHR3*, *TuE75*, *TuE78*, and *TuFtz-f1* (Figure 5C). For the vitellogenesis-related genes, the expressions of *TuVg1*, *TuVg2*, and *TuVgR1* were reduced by 40% (*p* = 0.0125), 50% (*p* = 0.0034), and 41% (*p* = 0.0041), respectively; for *TuVg3*, we also saw a small decrease in expression, but this was not significant (*p* = 0.0934) (Figure 5D).

## 3. Discussion

In insects, ecdysteroid hormone plays a vital role in reproduction by regulating the physiological process [3,5,7,37,38]. In *D. melanogaster*, ecdysteroid could regulate the germline stem cell increase induced by mating and egg production to sustain reproductive success response to the mating [39]. It also could regulate the timing of border-cell migration to affect ovary development [40]. The regulatory mechanism of ecdysteroid for the vitellogenin biosynthesis and oogenesis in other insects has been abundantly reported [41,42,43,44,45]. In Acari, observations of ecdysteroid hormone stimulating the synthesis of vitellogenin and being involved in oogenesis have also been reported in ticks [33,34,46,47]. However, little information on the regulatory function of the ecdysteroid hormone in the reproduction of spider mites is available.

In this study, it was interesting that we found that the expression patterns of the Halloween genes in the reproduction process of *T. urticae* were in accordance with the oviposition dynamic and the expression patterns of the vitellogenesis-related genes *TuVg* and *TuVgR.* In addition, in an RNAi assay, the knockdown of the Halloween gene *TuSpo* provoked a significant decrease in *TuVg* and *TuVgR* and also a significantly reduced oviposition. Therefore, we believe that our results demonstrated that the Halloween gene *TuSpo* is involved in the regulation of vitellogenin biosynthesis to govern the oviposition of the reproduction process in *T. urticae*. To a similar extent, in *S. gregaria* and *D. citri*, the expression levels of Halloween genes decreased after adult molting, then increased gradually during the female reproductive cycle [27,29,48]. In *V. destructor*, three Halloween genes, *Spo*, *Dib*, and *Shd*, were up-regulated during the reproductive stage [31]. These previous studies in insects are consistent with ours in spider mites. In insects, the RNAi-mediated knockdown of an ecdysteroid biosynthesis Halloween gene also could disrupt the expression of *Vg* or *VgR* and decrease number of eggs in the oviposition process [6,38,49,50,51]. Based on the abovementioned results in insects and mites, we believe that this approach can provide a new control strategy of important pests. Indeed, interference in the ecdysteroid signaling pathway can be performed by agonists and antagonists also of natural products from botanic origin [52,53,54,55]. Silencing the expression of *TuSpo* resulted in a severe decrease in oviposition. These results were similar to those reported in *S. gregaria*, where the downregulation of *Spo* resulted in shorter oocyte length, smaller eggs, and less hatching [27]. In *D. melanogaster*, the downregulation of *Shd* also resulted in a lower oviposition, but the effect was milder [28]. Based on previous experiments in insects and ticks together with our current results in spider mites, we believe that we can conclude that the ecdysteroid hormone plays a pivotal role in reproduction, specifically in the expression of the vitellogenesis-related genes. However, more investigations, for instance, on other regulatory (co-)factors and/or transcription factors such as as nuclear receptors [56], are necessary to better understand the regulatory cascade of vitellogenesis, which is the yolk protein formation in the oocytes, and the processes of germline formation, oogenesis, and choriogenesis in the female mite.

Hence, we hope that in the near future there may be an optimization of the analytical methodologies to determine the ecdysteroid hormone titer in the tiny organisms of spider mites. Today, advanced liquid chromatography/mass spectrometry (LC-MS/MS) technology could determine the ecdysteroid hormone (ponasterone A) in one sample of 500 mg of spider mites (i.e., mixed ages of nymphs and female adults) [18]. Such technology development is needed and will certainly stimulate more fundamental insights on the relationship between hormone titers, Halloween expression levels and the development and reproduction in spider mites and other tiny organisms in the clade of Ecdysozoa.

On the effect that we observed against the lifespan in ds*TuSpo*-injected female mites of *T. urticae*, we can, however, not yet explain the regulatory mechanism behind it. Indeed, it should be remarked here that, today, there is little to no information on this aspect. In some insects, a relationship has been reported between ecdysteroid and juvenile hormone (JH) and the insect lifespan [57,58,59,60]. We hope that future research will be performed to better understand the regulation of lifespan in spider mites as well.

In our RNAi experiment with dsTu*Spo* in *T. urticae*, we did not observe any effect on the offspring of the surviving females. We believe that this is most likely related to the amount of dsRNA injected, which was 60–80 ng per female in the current experiment. We expect that higher amounts of dsRNA can provide a clear transgenerational effect as has been seen in other insects. In *S. gregaria* and *N. lugens*, knockdown of Halloween genes resulted in a significant reduction of egg hatching [8,27]. Also, strong transgenerational effects have been reported in stinkbugs of *Euschistus heros* after dsRNA treatment of the female adults [61]. Hence, in a previous experiment with *T. urticae*, 2000 ng/µL of dsRNA was treated via a leaf disc and fed to spider mite females, and this treatment produced phenotypic effects in 3.6% of the offspring [62], confirming that a transgenerational activity by RNAi is possible in spider mites, but the effects depend on the concentration of dsRNA. Finally, we also believe that the modern technology of CRISPR/Cas9 has the potential to introduce new control technique based on essential genes, as has been seen in different pest insects, for instance, in relation to ecdysteroid hormone production and to realize sterility [63,64,65].

In conclusion, in this project with spider mites of *T. urticae*, the silencing of the Halloween gene *Spook* reduced the expression of the vitellogenin genes and their receptor gene, which led to a significant decrease in the egg production of female mites. These results illustrate that *Spook* plays a crucial role in regulating the oviposition in the reproduction process in *T. urticae*, as presented in Figure 6. We believe that this regulatory function may provide a new way to control spider mites of *T. urticae* and thereby avoid the rapid increase in resistance caused, in part, by the very high reproductive capacity of these important pest organisms. The new information from this project provides a better understanding of the ecdysteroid hormone activity in spider mites on the regulation of their reproduction; however, more research is still needed, as discussed above.

## 4. Materials and Methods

### 4.1. Culturing of Mites and Oviposition Statistics

The sensitive/wild mite strain was derived from the Institute of Entomology, Guizhou University, Guiyang, China and has been reared for more than ten years in-house and without exposure to pesticides. The mites were reared on beans (*Phaseolus vulgaris*) at 27 ± 1 °C, 65 ± 5% relative humidity, and a photoperiod of 14 h:10 h (L:D) in the laboratory as described before [66]. For the experiments, unfertilized adult female mites that molted for the last time in the same development period were screened for oviposition every 24 h until they stopped egg-laying. When there were no eggs for three consecutive days, the females were considered to have stopped laying eggs. These unfertilized adult females were not paired for the entire statistical period. One-way analysis of variance (ANOVA) with Tukey’s honestly significant difference (HSD) test was used to analyze the significant difference.

### 4.2. Expression Dynamics of the Halloween Genes in the Oviposition Period

To investigate the detailed expression dynamics of the Halloween genes (*TuSpo*, *TuDib*, *TuSad*, and *TuShd*) on different days, we collected adult female mite samples at different reproductive times at 24 h intervals starting from molting. The day that the mite had molted into the adult stage was named day 1. Every individual sample included 50 female mites, and four biological replicates were conducted. Total RNA of all samples was extracted using TRIzol (Sangon Biotech, Shanghai, China) according to the manufacturer’s protocol. The RNA integrity was checked on a 1% agarose gel, and its concentration was measured with a NanoDrop and Agilent 2100 bioanalyzer (Thermo Fisher Scientific, Waltham, MA, USA). StarScript II RT Mix with gDNA Remover (GenStar, Beijing, China) was used to synthesize the first-strand cDNA. Every cDNA was diluted 25-fold using RNase-free water and stored at −20 °C. The reverse transcription–quantitative polymerase chain reaction (RT-qPCR) specific primers were designed using the NCBI primer designing tool (https://www.ncbi.nlm.nih.gov/tools/primer-blast/, accessed on 11 April 2022). The 10.0 µL reaction system contained 5.0 µL of 2x RealStar Green Fast Mixture (GenStar, Beijing, China), 0.5 µL of forward primer, 0.5 µL of reverse primer, and 4.0 µL of cDNA. The reference gene *ATP* was used to normalize gene expression levels [66]. The relative expression levels were calculated using the method of 2^−ΔΔCT^. ANOVA with Tukey’s HSD test was used to analyze the significant difference in gene expressions in SPSS version 22.0 (IBM, Armonk, NY, USA).

### 4.3. RNAi of TuSpo

The dsRNA was synthesized in vitro using the Transcript Aid T7 High Yield Transcription Kit (Thermo Scientific, Shanghai, China) according to the manufacturer’s protocol. A total of 20 µL of the resulting transcripts was purified according to the kit instructions, to assure the quality of the synthesized dsRNA, and stored at −80 °C. The virgin adult female mites that had molted in the adult stage within 12 h were used for the RNAi experiment. Our injection method referred to previously existing injection methods [62,67,68]. Approximately 6–8 nL (concentration: 10 µg/µL) of dsRNA was injected into the mites. After injection, four biological replicates were performed to detect the silencing efficiency of Halloween gene by RT-qPCR at 24 h, and other female mites were assigned for phenotypic observation (egg-laying, oviposition cycle, and lifespan). The treatment of ds*egfp* was performed as a control. The oviposition was counted at 24 h intervals until the female mites died at day 10. Student’s *t*-tests were used to determine the significant differences between treatment and control groups (ds*egfp*). To evaluate the effects of dsRNA on the next generation, we analyzed the offspring development. In addition, the expressions of other Halloween genes, the ecdysteroid signaling pathway genes, and the vitellogenesis-related genes *Vg* and *VgR* were investigated.

## Figures and Tables

**Figure 1 ijms-24-14797-f001:**
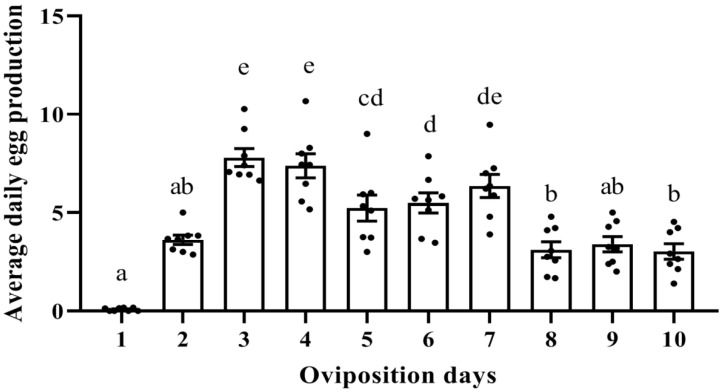
Average daily egg production (number of eggs per day and per mite) of female adult mites of *T. urticae* from the 1st day of the adult stage (=day 1) until death of the female mite at day 10. Every point represents a biological replicate and each replicate included 15 mites. In this experiment, we performed eight biological replicates per time point (day). The results are presented as mean (±SE) based on eight biological replicates per day. Lowercase letters above each column indicate significant differences amongst the ten days using one-way analysis of variance (ANOVA) followed by Tukey’s honestly significant difference (HSD) test (*p* < 0.05).

**Figure 2 ijms-24-14797-f002:**
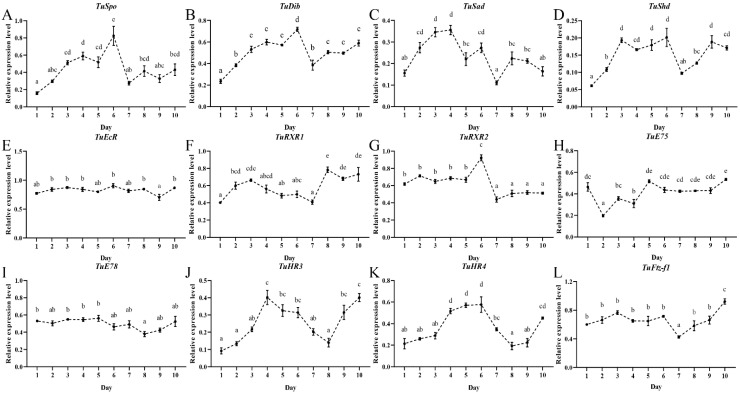
Expression pattern of the ecdysteroid biosynthesis Halloween genes (**A**–**D**) and the ecdysteroid signaling pathway genes (**E**–**L**) in female mites of *T. urticae* as determined by quantitative real-time PCR (RT-qPCR). The relative expression levels were calculated using the method of 2^−ΔΔCT^ and based on the value of the lowest expression level. Lowercase letters above each bar indicate significant differences amongst the ten days using one-way ANOVA followed by a Tukey’s HSD test (*p* < 0.05).

**Figure 3 ijms-24-14797-f003:**
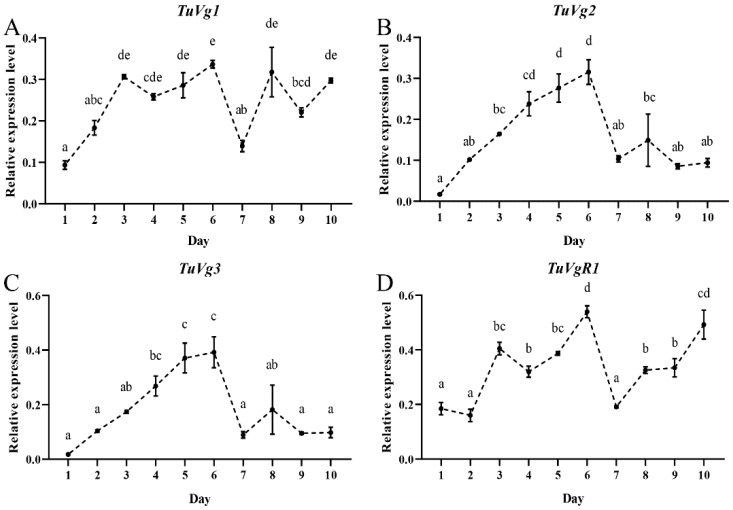
Expression pattern of the vitellogenesis-related genes, *Vg* (**A**–**C**) and *VgR* (**D**), in female mites of *T. urticae* as determined RT-qPCR. The relative expression levels were calculated using the method of 2^−ΔΔCT^ and based on the value of the lowest expression level. Lowercase letters above each bar indicate significant differences amongst the ten days using ANOVA followed by a Tukey’s HSD test (*p* < 0.05).

**Figure 4 ijms-24-14797-f004:**
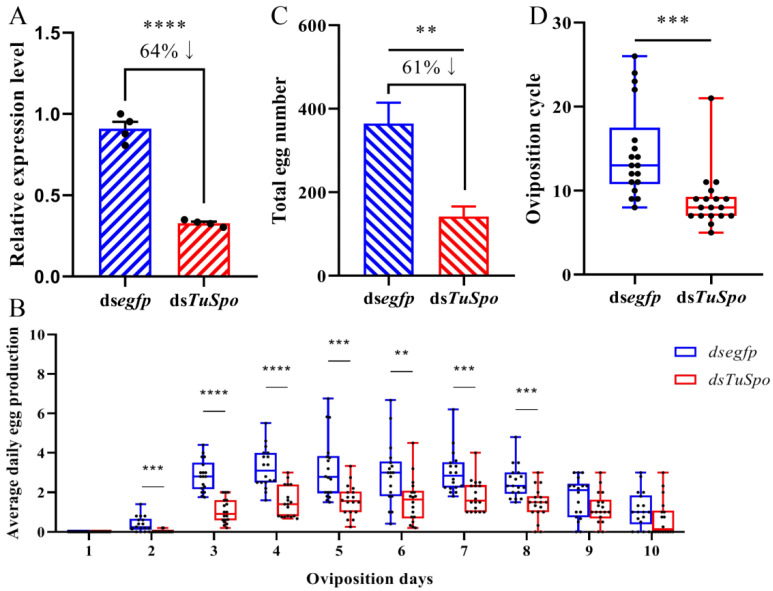
RNAi effects against *TuSpo* by injection in female mites of *T. urticae*. (**A**) The silencing efficiency of *TuSpo* was detected by RT-qPCR at 24 h after the female mites were injected with 60–80 ng of ds*TuSpo*. The mean (±SE) expression level is based on four biological replicates, and each replicate consisted of 50 mites. The relative expression was calculated based on the value of ds*efgp* (control). The down arrow indicates a decrease of 64% in relative expression of *TuSpo*. (**B**) Effects of ds*TuSpo* injection in female mites against the average daily egg production. Every point represented a small round leaf (2 cm diameter) with 5 mites on each leaf. The mean (±SE) average daily egg production is based on 18 biological replicates. (**C**) Effects of ds*TuSpo* injection in female mites on the total egg number. The mean (±SE) total egg number is based on four biological replicates, and we used 25 female mites per replicate. The down arrow indicates a decrease of 61% in total egg number. (**D**) Effects of ds*TuSpo* injection in female mites on the oviposition cycle. The mean (±SE) oviposition period is based on 18 biological replicates, and we used 5 mites per replicate. The significant difference between the two groups is indicated with ‘**’, *p* < 0.01, ‘***’, *p* < 0.001 and ‘****’, *p* < 0.0001 after a Student’s *t*-test.

**Figure 5 ijms-24-14797-f005:**
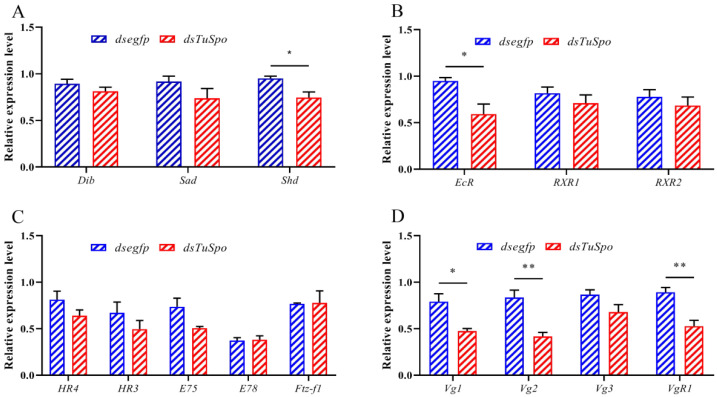
Relative expression levels of the ecdysteroid biosynthesis and signaling pathway genes and the vitellogenesis-related genes in female mites of *T. urticae*, as determined by RT-qPCR at 24 h after the female mites were injected with 60–80 ng of ds*TuSpo*. (**A**) Relative expression levels of the other ecdysteroid biosynthesis Halloween genes *TuDib*, *TuSad*, and *TuShd*. (**B**) Relative expression levels of the ecdysteroid receptor complex genes *TuEcR*, *TuRXR1*, and *TuRXR2*. (**C**) Relative expression levels of the ecdysteroid signaling genes *TuHR4*, *TuHR3*, *TuE75*, *TuE78*, and *TuFTZ-f1*. (**D**) Relative expression the vitellogenin genes and their receptor gene, *TuVg1*, *TuVg2*, *TuVg3*, and *TuVgR1*. The mean (±SE) expression level is based on four biological replicates, and we used 50 mites per replicate. Significant differences between the treatment and control are indicated with ‘*’, *p* < 0.05 and ‘**’, *p* < 0.01 after a Student’s *t*-test.

**Figure 6 ijms-24-14797-f006:**
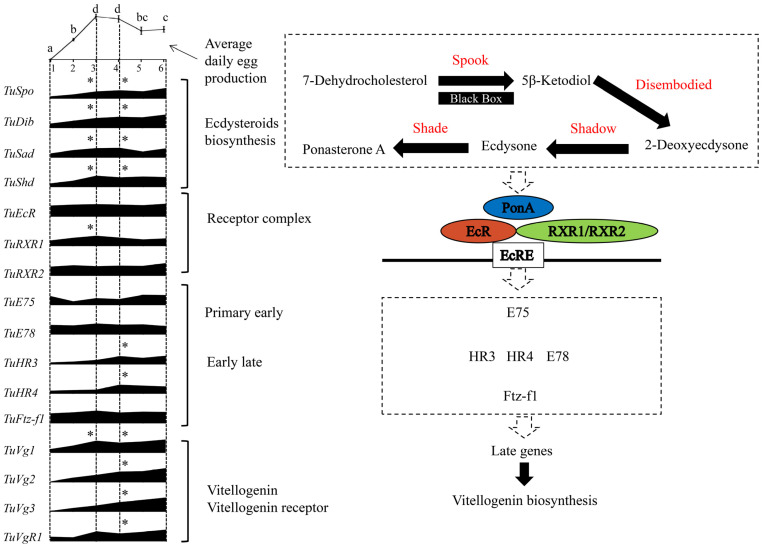
Scheme proposing the ecdysteroid biosynthesis and signaling cascade pathway regulating vitellogenesis and reproduction in *T. urticae*. Lowercase letters above the average daily egg production indicate significant differences during the first six days (day 1–6) in the adult stage using ANOVA followed by a Tukey’s HSD test (*p* < 0.05). ‘*’ indicates a significant difference of change during day 1–6 in the adult stage for the relative expression levels of the respective gene. Data on egg production and expression of genes are based on Figure 1, Figure 2 and Figure 3.

## Data Availability

Data are contained within the article or Appendix A.

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
