# Peer review of "Ecdysteroid Biosynthesis Halloween Gene Spook Plays an Important Role in the Oviposition Process of Spider Mite, Tetranychus urticae"

_ijms, 2023, doi:10.3390/ijms241914797_

Round 1

Reviewer 1 Report

The study is well written but I have somme comments. These are:

please consider the medhodology as second section in the study

Line 234: How unfertilized females were monitored until they stopped egg laying? Or did you start experiments using the offspring from from these females or did you paired the experimental female mites?

Regarding the strain, how long did you rear them at your lab?

Line 241: Add the number of biological days of the molting day of the mite. For example, the day the egg was hatched: day 0

Results: 

The term of spawning is used mostly for fish. Please use oviposition period instead of it

Figure 1: How much replicate do you have, please eloborate it in methodology

Discussion

Line 191: present your highlighted results at the beginning

Line 189: reformulate the sentence 

I have recommendations in other section but any case provide a prof-read

Author Response

Thank you very much for taking the time to review this manuscript. Please find the detailed responses below.

Reviewer 2 Report

Dear Authors,

              I am very glad to have had the opportunity to review your work. The problem with spider mites is a huge problem in vegetables, orchards, ornamental plants and field plants. Combating mites is a challenge for agriculture due to: the pests becoming resistant to chemicals, the problem of high reproduction and easy and quick spread to entire plants and crops, and poor spraying techniques. Hence, the research conducted by the authors gives hope for counteracting the effects of spider mite resistance to chemical preparations, after knowing the genes that influence their rapid reproduction.

            The aim of the work is very well formulated. The methodological part is described very carefully and the reader must find it. I suggest placing it after the Introduction. The methods of statistical analysis should be described in as much detail as possible, and not just a demonstration that an analysis of variance was performed. This part lacks a clear presentation of the research methodology. In the conclusions, it might be worth describing how the experiments on the Halloween gene TuSpo will be used in practice in the context of reducing egg reproduction. The ecdysteroid biosynthetic pathway The Halloween TuSpo gene plays a key role regulating egg laying during reproduction in T. urcitae. Using this fact when looking for compounds to combat spider mites that inhibit this gene.

               Very interesting and scientifically high-level work. After responding to the comment, it is ready for printing.

Kind regards,

Author Response

(The authors gave the same response as above.)

Reviewer 3 Report

General comments

I have read the manuscript (ijms-2595460), Entitled: Ecdysteroid biosynthesis Halloween gene Spook plays an important role in the oviposition process of spider mite, Tetranychus urticae, written by Wang et al. for publication in MDPI/IJMS. I found the manuscript nicely written and suitable for publication after minor revision.

Line 231: T. urticae, Write all scientific name in italic font.

Did the author use a parametric test before ANOVA?

Check the manuscript title. Please follow the format uniformly.

Conclusion: The abstract and conclusion should not be repetitive. I would love to read striking points and take-home messages that will linger in the readers’ minds. What is the novelty, how does the study elucidate some questions in this field, and what contributions may the paper offer to the scientific community?

References: Please double-check the citations, their style, scientific name, spell check, and other grammatical errors.

Good Luck!

Author Response

(The authors gave the same response as above.)
